# Presentation, Management, and Outcome of Tick-Borne Encephalitis in Patients Referred to Infectious Diseases or Neurology [note 1]

**DOI:** 10.3390/jcm14010045

**Published:** 2024-12-25

**Authors:** Jana Gulin, Lučka Marija Neudauer, Nataša Kejžar, Fajko F. Bajrović, Stefan Collinet-Adler, Daša Stupica

**Affiliations:** 1Department of Infectious Diseases, University Medical Centre Ljubljana, 1000 Ljubljana, Slovenia; jana.gulin@kclj.si; 2Faculty of Medicine, University of Ljubljana, 1000 Ljubljana, Slovenia; 3Department of Dermatology, University Medical Centre Ljubljana, 1000 Ljubljana, Slovenia; lucka.marija.neudauer@kclj.si; 4Institute for Biostatistics and Medical Informatics, Faculty of Medicine, University of Ljubljana, 1000 Ljubljana, Slovenia; natasa.kejzar@mf.uni-lj.si; 5Department of Neurology, University Medical Centre Ljubljana, 1000 Ljubljana, Slovenia; fajko.bajrovic@kclj.si; 6Institute of Pathophysiology, Faculty of Medicine, University of Ljubljana, 1000 Ljubljana, Slovenia; 7Department of Infectious Diseases, Park Nicollet/Health Partners, Methodist Hospital, Saint Louis Park, MN 55426, USA; stefan.collinet_adler@parknicollet.com; 8Department of Infectious Diseases, Faculty of Medicine, University of Ljubljana, 1000 Ljubljana, Slovenia

**Keywords:** tick-borne encephalitis, clinical management, neuroimaging, outcome, lumbar puncture

## Abstract

**Background**: In Slovenia, patients with suspected tick-borne encephalitis (TBE) were historically referred to infectious diseases (ID), but during the COVID-19 pandemic, there were increased referrals to neurology. This study compared the clinical management of TBE patients between ID specialists and neurologists and assessed patients’ outcomes. **Methods**: We retrospectively reviewed the clinical, laboratory, and imaging data of 318 adult patients with TBE managed by ID (n = 256; 80.5%) and neurology (n = 62; 19.5%) at a tertiary centre in Slovenia between March 2020 and September 2022 to explore variations in diagnostic and therapeutic approaches by specialty and to assess the severity and outcome of acute illness. **Results**: Patients referred to ID or neurology did not differ regarding their basic demographic and epidemiologic characteristics or basic laboratory parameters. However, patients referred to neurology more often presented with severe illness, including impaired consciousness and/or focal neurological signs (72.6% vs. 55.5%; *p* < 0.001). ID specialists used head imaging before lumbar puncture (6.6% vs. 64.5%; *p* < 0.001), performed microbiological tests other than for TBE (16.0% vs. 51.6%; *p* < 0.001), and empirically prescribed antimicrobials less often than neurology (5.1% vs. 22.6%; *p* < 0.001). When adjusting for age, sex, comorbidities, vaccination status, and the severity of acute illness, clinical outcomes were similar between the two groups of patients, but those with more severe acute illness had higher odds for incomplete recovery. **Conclusions**: Differences in clinical presentation between ID and neurology referrals could only partially explain the narrower diagnostic and therapeutic approach used by ID, which, given the study design, was not associated with adverse outcomes. Additionally, in patients with clinical characteristics suggestive of TBE in endemic areas, tremor in the absence of other focal neurological signs or impaired consciousness may not necessitate head imaging before lumbar puncture. Future prospective studies could help to optimise the management of this clinical syndrome.

## 1. Introduction

Acute community-acquired central nervous system (CNS) infections, such as encephalitis, meningitis, and meningoencephalitis, represent a major clinical challenge due to diverse aetiologies and potential for severe outcomes. Clinical guidelines on the management of acute community-acquired CNS infections recommend that clinical presentation, basic laboratory blood and cerebrospinal fluid (CSF) analyses, specifics of local epidemiology, and individual host risk factors should guide the selection of diagnostics and therapeutics [1,2,3]. Across Europe, minor variations exist in clinical guidelines for managing community-acquired CNS infections, including viral encephalitis/meningoencephalitis [4]. There is a need for an update of the 2008 Infectious Diseases Society of America guidelines for the management of encephalitis [5]. Depending on local medical culture, patients with encephalitis/meningoencephalitis may be managed by infectious diseases (ID) specialists and/or neurologists. Interdisciplinary management is preferable [4,5]. Improved training of physicians in the diagnosis and management of encephalitis may help optimise clinical management of these patients [5,6].

Slovenia has one of the highest incidence rates of tick-borne encephalitis (TBE) and Lyme borreliosis (LB) in Europe [7,8]. Both illnesses are transmitted by ticks and may present as CNS infections [9,10]. TBE virus is the most prevalent virus, and *Borrelia burgdorferi* sensu lato is the most prevalent bacteria causing aetiologically identified acute community-acquired CNS infections in adults in Slovenia [11]. Although there is no known specific treatment for TBE [9], other aetiologies with effective specific treatment could potentially present in a similar fashion to TBE. Clinical outcomes in patients with TBE treated with adequate supportive care alone should be the same even if the diagnosis is missed, but there is potential for harm from overly broad and lengthy empiric testing and treatment. Securing a diagnosis of TBE and excluding other aetiologies quickly and efficiently permits the medical team to streamline management and deliver cost-effective care.

At our centre, patients suspected of TBE were historically referred to ID, but for organisational reasons during the COVID-19 pandemic, they were also referred to neurology. This change in referral pattern provided an opportunity to compare and contrast the evaluation and management of patients with syndromes that were ultimately found to be due to TBE by two different departments. We aimed to assess whether there was a difference in approach between the two departments, and, if so, whether there was an advantage to using one approach over the other. These findings might inform future guidelines and improve interdisciplinary management strategies for CNS infections, particularly in regions with high tick-borne disease burden like Slovenia.

To this end, we analyzed the clinical presentation at admission, diagnostic and therapeutic management, and the outcome of patients with TBE initially referred to ID or neurology. We also explored associations between the severity of acute illness and clinical outcome.

## 2. Materials and Methods

### 2.1. Setting and Patients

In this retrospective observational cohort study, we reviewed medical charts, including the medical histories of patients ≥ 18 years old discharged with a diagnosis of TBE from the ID or neurology departments at the University Medical Centre Ljubljana, Slovenia, between March 2020 and September 2022. TBE was defined according to European criteria: symptoms/signs of meningitis and/or encephalitis, and/or radiculitis/myelitis, cerebrospinal fluid (CSF) pleocytosis (>5 × 10^6^ cells/L), and demonstration of TBE virus specific IgM and IgG antibodies in serum. In patients previously vaccinated against TBE, we evaluated the intrathecal synthesis of specific antibodies [9]. STROBE guidelines for reporting were followed. Potential confounding could be present due to unmeasured or unreported variables inherent in the retrospective design of the study.

### 2.2. Evaluation of Patients

The first 48 h after admission were considered the period of initial management because in our institution, serologic test results for TBE virus are available within 24–48 h. The severity of acute illness at the time of admission and upon hospital discharge were defined as mild, moderate, or severe [12]. To provide a more detailed quantitative assessment of the severity of acute illness, a composite clinical severity score was developed, consisting of 16 clinical symptoms, and signs were weighted using severity points so that cumulative scores of 1–4 would correspond to mild, 5–8 to moderate, and ≥9 to severe acute illness (Table 1).

TBE outcome was assessed using the modified Rankin scale [13] at hospital discharge and follow-up visits approximately 6 months after hospital admission. Incomplete recovery was defined as a modified Rankin score ≥ 2.

### 2.3. Laboratory Evaluation

IgM and IgG antibodies to TBE virus were measured using the Enzygnost^®^ Anti-TBE Virus test (SiemensGmbH, Marburg, Germany). Results were interpreted according to the manufacturer’s instructions. Intrathecal synthesis of TBE antibodies was calculated according to the method described by Reiber and Peter [14]. Antibody index values > 1.4 were considered indicative of intrathecal production of TBE antibodies.

### 2.4. Statistical Analysis 

The patients’ baseline characteristics, indications for head imaging and microbiological testing, were described by frequency (percentage) for categorical variables and using medians (IQR) for numerical variables. Comparisons of the studied groups were performed with two-tailed Pearson’s χ^2^ or Fisher exact test for categorical variables, and with Wilcoxon rank sum tests for continuous variables. Due to multiple comparisons, *p*-values < 0.001 were considered statistically significant. The association between admitting service and the decision to perform head CT before LP controlling for selected characteristics expected to influence this decision as well as the association between the admitting service and the clinical outcome at discharge and 6-month follow-up, controlling for the covariates expected to influence the outcome, were estimated using multiple logistic regression. The results are presented as odds ratios (ORs) with 95% confidence intervals (CIs).

## 3. Results

Of the 352 patients discharged with a diagnosis of TBE during the study period, 318 (90.3%) were included in the analysis: 256 (80.5%) patients were initially managed in ID and 62 (19.5%) in neurology (Figure 1).

### 3.1. Patients’ Characteristics at Admission

Patients initially referred to ID or neurology did not significantly differ in terms of basic demographic and epidemiologic characteristics or basic blood and CSF laboratory parameters (Table 2).

However, patients referred to ID more often reported fever, while those referred to neurology more often had impaired consciousness or focal neurological signs. Those referred to neurology included a higher proportion of patients presenting with meningoencephalitis or meningoencephalomyelitis and had more severe acute illness as assessed by the composite clinical score at admission (Table 2). Although not statistically significant, patients seen in ID more frequently presented with a combination of ≥3 clinical characteristics suggestive of TBE (recent tick bite, biphasic course, fever, headache, nausea/vomiting, nuchal rigidity, tremor) than those seen in neurology (225/256; 87.9% vs. 45/62; 72.6%; *p* = 0.005).

### 3.2. Initial Head Imaging According to Admitting Service

Computed tomography (CT) of the head was performed before lumbar puncture (LP) in 57/318 (17.9%) patients (Table 3) with a significantly higher proportion in neurology (40/62; 64.5%) than in ID (17/256; 6.6%); *p* < 0.001 (Table 3). In none of these cases did the CT results preclude LP.

Irrespective of whether CT was performed before LP or not, no post-LP complications were detected, apart from possible post-puncture headaches, which could not definitely be distinguished from TBE-associated headache.

Among 113 patients with isolated tremor at admission (without any other focal neurological signs, seizures, immunodeficiencies or impaired consciousness), CT was performed before LP in 12 (10.6%) cases, with a higher proportion in neurology (8/14, 57.1%) than in ID (4/99, 4.0%) (*p* < 0.001). In patients presenting with a combination of ≥3 clinical characteristics suggestive of TBE, the proportion of those with CT conducted prior to LP was lower in ID compared to neurology (13/225; 5.8% vs. 28/45; 62.2%; *p* < 0.001). None of these CT scans revealed pathological findings related to the clinical presentation that required additional treatment or diagnostics.

The logistic regression model showed that patients admitted in neurology were more likely to have had head imaging conducted before LP than those admitted in ID when controlling for the presence of focal neurological signs, tremor, ≥3 clinical characteristics suggestive of TBE, and severity of acute illness at hospital admission (Table 4).

### 3.3. Initial Microbiologic Diagnostics and Therapeutic Management According to Admitting Service

Among patients who were unvaccinated against TBE, the proportion of those who were additionally tested for TBE virus antibodies in CSF was lower in ID than neurology (12/236; 5.1% vs. 21/57; 36.8%; *p* < 0.001). Microbiologic testing other than serological tests for TBE virus and Lyme borreliae were performed less often in ID than neurology (Table 5).

Patients admitted to ID received empirical antimicrobial therapy less frequently than those admitted to neurology (13/256; 5.1% vs. 14/62; 22.6%; *p* < 0.001). Antibiotic therapy was prescribed in 8/256 (3.1%) patients in ID and 7/62 (11.3%) in neurology (*p* = 0.014), and antiviral therapy was started in 10/256 (3.9%) patients in ID and 13/62 (21.0%) in neurology (*p* < 0.001). Antibiotic therapy for possible or proven Lyme neuroborreliosis was prescribed in 32/256 (12.5%) patients in ID and 5/62 (8.1%) in neurology (*p* = 0.386).

### 3.4. Clinical Outcome According to Specialty Service

After initial management, 25/318 (7.9%) patients were transferred between departments due to logistics (Figure 1): the 18 patients transferred from neurology to ID tended to have more severe illness at hospital discharge than the 7 patients transferred from ID to neurology, though this difference was not statistically significant (median severity score 12.5; IQR 8.5–40.0 vs. median 7; IQR 6–11; *p* = 0.179). There was no significant difference in duration of hospitalisation (median 7 days; IQR 5–10 days vs. median 7 days; IQR 6–11 days; *p* = 0.148) or in severity of acute illness as assessed at hospital discharge (median 7; IQR 6–11 vs. median 7; IQR 6–11; *p* = 0.991) between patients discharged from ID versus neurology.

The clinical outcomes of TBE, as assessed by the modified Rankin score, at hospital discharge and at the 6-month follow-up in patients discharged from ID or neurology were comparable (at hospital discharge: median 1; IQR 1–2 vs. median 1; IQR 1–2; *p* = 0.982; at 6 months: median 0; IQR 0–1 vs. median 1; IQR 0–1; *p* = 0.082). Regardless of the discharging department, the modified Rankin score was higher at hospital discharge than at the 6-month follow-up (*p* < 0.001 for both comparisons).

The logistic regression model, adjusting for age, sex, Charlson comorbidity index, vaccination status, and severity of acute illness at hospital discharge, showed no significant differences in clinical outcomes between patients initially managed in neurology and those managed in ID (Table 6). More severe acute illness was associated with higher odds of incomplete recovery at hospital discharge (modified Rankin score ≥ 2).

## 4. Discussion

This retrospective study of TBE patients managed by two separate specialties in the same university centre showed that ID performed fewer diagnostics and prescribed empirical antimicrobials less often than neurology. These differences could only be partially explained by differences in clinical presentation. The narrower initial diagnostic and therapeutic management used by ID was not associated with adverse events, duration of hospitalisation, or clinical outcome at hospital discharge or at the 6-month follow-up. This is not surprising since there is no specific treatment for TBE and only patients with a known diagnosis of TBE were included in the study. The severity of acute illness, as assessed at hospital discharge, was the main predictor of clinical outcome, irrespective of the admitting specialty.

To our knowledge, this is the first study to directly assess differences in referral patterns and management strategies for patients with TBE between specialties. In the three years preceding the study period, 210 patients with TBE were discharged from ID, only five from neurology, supporting our assumption that the COVID-19 pandemic modified referral patterns of TBE patients within our institution. This shift provided a unique opportunity to investigate management differences between the two specialties. There is no reason to believe that these variations in practice are specific to the specialties themselves; rather, they may reflect the idiosyncrasies of our institution and our referral patterns.

At admission, patients referred to neurology had more severe illness, marked by a higher frequency of impaired consciousness and focal neurological signs. This could reflect the neurologists’ expertise in recognising and documenting neurologic impairment. However, even after adjusting for more subtle neurological signs (e.g., dysphasia, dysarthria, and ataxia), the difference in severity persisted, suggesting that general practitioners may be more likely to refer patients with more severe neurological symptoms to neurology, while those with milder symptoms are referred to ID. The cautious and broader evaluation and management by neurology is understandable in the context of severe illness with multiple possible aetiologies.

In clinical guidelines of management of encephalitis or viral meningoencephalitis [1,2,3], the timing of LP in relation to neuroimaging is addressed in various ways. Some guidelines do not specify the timing of LP relative to neuroimaging [1], while others recommend that LP should only be delayed under unusual circumstances that are not further specified [3]. More specific recommendations indicate that neuroimaging should be performed prior to LP in patients presenting with papilledema, altered consciousness, a focal neurological deficit, or new onset seizures [2,16]. This aligns with the ESCMID guidelines for the management of acute bacterial meningitis [15], which emphasise the importance of neuroimaging before LP in patients with signs of increased intracranial pressure or other neurological abnormalities. Our study found that CT imaging prior to LP was performed significantly more often in patients admitted to neurology (40/62, 64.5%) than in those admitted to ID (17/256; 6.6%). This difference may be attributed to variations in clinical presentation, as patients with more severe illness, including focal neurological signs and impaired consciousness, were more likely to be referred to neurology. According to the guidelines [2,15], head imaging before LP should have been performed in 140/256 (54.7%) and 43/62 (69.4%) patients admitted to ID and neurology, respectively. However, in our study, clinicians did not adhere to head imaging recommendations in 135/256 (52.7%) patients in ID and 27/62 (43.5%) patients in neurology. A head CT was performed without an indication in 6/116 (5.2%) patients in ID and in 12/19 (63.2%) in neurology and omitted when indicated in 129/140 (92.1%) patients in ID and 15/43 (34.9%) in neurology.

For comparison, studies from the United States and the Netherlands found noncompliance with cranial imaging recommendations in patients with community-acquired meningitis ranging from 43% to 48% [17,18,19], primarily due to performing CT imaging when not indicated (64% (355/549) patients in the US study and 67% (526/781) patients in the Dutch study) [19]. In our study, there was a higher proportion of patients in ID (92%) without head CT conducted before LP, i.e., when indicated by guidelines, compared to neurology (35%), and in comparison to the 6% to 17% noncompliance reported in previous studies of patients with community-acquired meningitis [18,19], this may suggest that ID specialists, with their extensive clinical experience in managing TBE in an endemic setting, tend to rely more on clinical judgment than guideline recommendations. Additionally, the lower proportion of patients in ID (5%) with head CT, when not indicated per guideline recommendations, than those in neurology (63%) and in previous studies on patients with community-acquired meningitis (64% to 67%) [18,19] may be due to our ID physicians adopting a less defensive approach when deciding on head imaging prior to LP in patients suspected of TBE. As expected, given the study design and inclusion criteria, no adverse events were associated with omitting head imaging prior to LP, and no significant additional pathology was found on head imaging. We could not evaluate the general appropriateness of this diagnostic approach since patients were analysed only after a TBE diagnosis was secured.

As suggested by Salazar et al., the probability of significant cerebral mass is very low in patients with acute symptoms consistent with community-acquired meningitis and impaired mental status but without focal neurologic deficits [18]. Based on the results of our study, it seems reasonable to propose that isolated new-onset tremor, in the absence of other focal neurological signs or impaired consciousness, may not require neuroimaging before LP in patients suspected of TBE. Tremor, in this context, is unlikely to indicate increased intracranial pressure due to a cerebral mass or diffuse brain oedema, which would typically contra-indicate an LP [15,20]. To validate this hypothesis and diagnostic approach, further prospective studies are warranted in patients suspected of TBE.

The narrower microbiological testing and less frequent use of empirical antimicrobial therapy by ID specialists may be partially explained by the higher proportion of patients with a clinical syndrome suggestive of TBE referred to ID (87.9%) when compared to neurology (72.6%). However, even in patients with a clinical syndrome suggestive of TBE, ID specialists performed additional microbiologic testing less often (30/225; 13.3% vs. 19/45; 42.2%; *p* < 0.001) and prescribed empirical antibiotics less frequently (9/225; 4.0% vs. 9/45; 20.0%; *p* < 0.001) than neurologists. This suggests that local epidemiology and clinical experience with TBE in endemic areas may have played a significant role in decision making, with ID specialists at our centre feeling more confident in their initial suspected diagnosis and less inclined to pursue broad evaluations and treatments.

The severity of acute illness in TBE has been found to be the main predictor of clinical outcome [12,21,22]. The severity of TBE at presentation was higher in patients admitted to neurology than ID, but there were no significant differences found in clinical outcome at hospital discharge or at the 6-month follow-up between the two services. Possible explanations for this include potential variations in expression and duration of illness at the time of hospital admission and the fact that patients who were transferred from neurology to ID had a non-significant trend towards higher-than-median scores of illness severity. There also could have been unmeasured differences in supportive care between the two departments. Additionally, patients with undiagnosed co-infections may have benefitted from empiric broader treatment by neurology.

This study has several limitations. Major limitations include the retrospective nature of this study, leading to significant potential for unrecognised biases, as well as the limited study time period and small sample size. Patients were only included in the study after the diagnosis of TBE was secured, artificially favouring a more targeted work-up and conservative management. It is possible that patients initially presenting with greater diagnostic ambiguity, potentially missed by the analytical approach we used, were more often hospitalised in neurology. Additionally, our results may not apply to other geographical areas with different aetiologic spectra of encephalitis/meningoencephalitis [23].

## 5. Conclusions

Our study enabled a comparison between the management of TBE patients by two separate departments in a university hospital. At our centre, ID specialists used a narrower diagnostic and therapeutic approach in managing TBE than neurologists. These differences could only be partially explained by the preferential referral of patients with impaired consciousness and/or focal neurological signs to neurology. As might be expected given the inclusion criteria and lack of specific therapy for TBE, a narrower evaluation and management was not clearly associated with adverse outcomes. Isolated new-onset tremor may not be an indicator of increased intracranial pressure necessitating head imaging before LP in patients with clinical characteristics suggestive of TBE in endemic areas. These data may encourage prospective studies aiming to optimise diagnostic stewardship in managing these patients

## Figures and Tables

**Figure 1 jcm-14-00045-f001:**
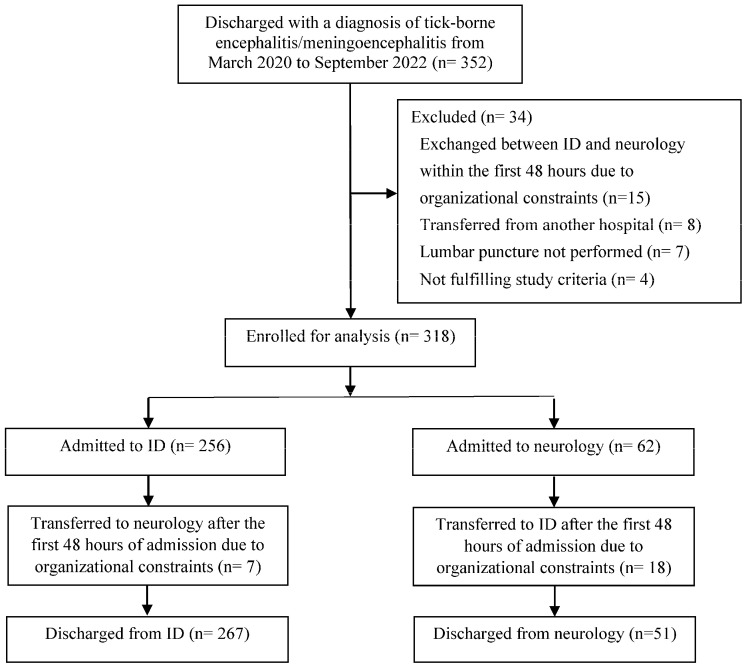
Study diagram.

**Table 1 jcm-14-00045-t001:** Clinical assessment (**A**) and composite clinical score (**B**) of the severity of tick-borne encephalitis at presentation and hospital discharge.

**A. Clinical Assessment of the Severity of Acute Illness**	
Subjective symptoms and objective signs	Severity ^1^
Symptoms/signs of meningeal involvement including fever, headache, nausea/vomiting, and/or nuchal rigidity	Mild
Monofocal neurological signs and/or moderate diffuse brain dysfunction manifesting as moderate quantitative impaired consciousness (somnolence) and/or any qualitative consciousness impairment, such as slowness, disorientation, confusion, agitation, memory, or concentration impairment	Moderate
Multifocal neurological signs and/or severe diffuse brain dysfunction manifesting as severe quantitative impaired consciousness (stupor or coma) and/or a combination of a monofocal neurological sign and moderate quantitative or qualitative consciousness impairment and/or a combination of quantitative and qualitative consciousness impairment	Severe
**B. Composite clinical score for the assessment of the severity of acute illness**	
Subjective symptoms and objective signs	Points ^2^
Fever ≥ 37.5 °C	0/1
Headache	0/1
Nausea or vomiting	0/1
Nuchal rigidity	0/1
Dysphasia	0/4
Dysarthria	0/4
Respiratory failure	0/8
Cranial nerve impairment	
One nerve	0/4
Two or more nerves	0/8
Upper or lower limb palsy	
One limb	0/4
Two or more limbs	0/8
Sensory disturbance	0/4
Sphincter dysfunction	0/4
Tremor	0/4
Ataxia	0/4
Seizure	0/4
Qualitative consciousness impairment ^3^	0/4
Quantitative consciousness impairment	
Somnolence (arouses to minor stimulation/voice)	0/4
Stupor (arouses to repeated stimulation or pain)	0/6
Coma ^4^ (unresponsive)	0/8

^1^ Symptoms/signs scored as 0 = absent or 1 to 8 = present such that cumulative scores of 1–4 would correspond to mild, 5–8 to moderate, and ≥9 to severe acute illness. ^2^ Maximum total score is 68. ^3^ Such as confusion, disorientation, slowness, agitation, memory, or impaired concentration. ^4^ For comatose patients, maximal score was automatically assigned for dysphasia, dysarthria, cranial nerve impairment, limb palsy, sensory disturbance, sphincter dysfunction, and qualitative consciousness impairment.

**Table 2 jcm-14-00045-t002:** Demographic, clinical, and laboratory characteristics of patients with tick-borne encephalitis at hospital admission according to referral to the infectious diseases department or the neurology department.

Characteristic	All *n* = 318	ID *n* = 256	Neurology *n* = 62	*p*-Value ^1^
Age	57 (42–66)	57 (41–65)	58 (44–67)	0.510
Male sex	184 (57.9)	149 (58.2)	35 (56.5)	0.802
Charlson comorbidity index	1 (0–2)	1 (0–2)	1.5 (0–3)	0.647
Severely immunocompromised ^2^	2 (0.6)	0 (0)	2 (3.2)	0.038
Vaccinated against TBE	25 (7.9)	21 (8.2)	4 (6.5)	0.796
Tick bite	226/302 (74.8)	190/250 (76.0)	36/52 (69.2)	0.306
Duration of symptoms/signs	10 (6–16)	11 (6–17)	7 (5–14)	0.043
Biphasic course of illness	156 (49.1)	132 (51.6)	24 (38.7)	0.069
Clinical signs/symptoms				
Fever	283 (89.0)	240 (93.8)	43 (69.4)	**<0.001**
Headache	216 (67.9)	170 (66.4)	46 (74.2)	0.239
Nausea or vomiting	39 (12.3)	28 (10.9)	11 (17.7)	0.143
Nuchal rigidity	139 (43.7)	117 (45.7)	22 (35.5)	0.146
Focal neurological signs ^2^	167 (52.5)	128 (50.0)	39 (62.9)	**<0.001**
Dysphasia/dysarthria	20 (6.3)	9 (3.5)	11 (17.7)	**<0.001**
Cranial nerve impairment	4 (1.3)	0	4 (6.5)	0.001
Upper or lower limb palsy	15 (4.7)	4 (1.6)	11 (17.7)	**<0.001**
Tremor	141 (44.3)	116 (45.3)	25 (40.3)	0.478
Ataxia	10 (3.1)	4 (1.6)	6 (9.7)	0.005
Focal neurological signs without tremor	26 (8.2)	12 (4.7)	14 (22.6)	**<0.001**
Seizure ^2^	2 (0.6)	1 (0.4)	1 (1.6)	0.352
Impaired consciousness ^2^	57 (17.9)	36 (14.1)	21 (33.9)	**<0.001**
Qualitative	54 (17.0)	33 (12.9)	21 (33.9)	**<0.001**
Quantitative	16 (5.0)	9 (3.5)	7 (11.3)	0.020
Glasgow comma scale of <10 ^2^	5 (1.6)	4 (1.6)	1 (1.6)	>0.999
Clinical presentation of TBE				**<0.001**
Meningitis	131 (41.2)	114 (44.5)	17 (27.4)	
Meningoencephalitis	173 (54.4)	138 (53.9)	35 (56.5)	
Myelitis	14 (4.4)	4 (1.6)	10 (16.4)	
Clinical severity of illness				**<0.001**
Mild	131 (41.2)	114 (44.5)	17 (27.4)	
Moderate	139 (43.7)	117 (45.7)	22 (35.5)	
Severe	48 (15.1)	25 (9.8)	23 (37.1)	
Clinical severity score of illness at admission	5 (2–7)	5 (2–7)	6 (3.3–11.8)	0.001
Criteria for head imaging prior to LP ^2^	183 (57.5)	140 (54.7)	43 (69.4)	0.045
CSF				
Leukocyte count (×10^6^/L)	78 (43–161)	77 (42–148)	81 (44–187)	0.485
Polymorphonuclear count (×10^6^/L)	18 (7–37)	18 (7–37)	21 (10–53)	0.155
Protein concentration (mg/mL)	0.88 (0.69–1.14)	0.88 (0.69–1.13)	0.89 (0.74–1.23)	0.340
Glucose concentration (mmol/L)	3.0 (2.7–3.3)	3.0 (2.7–3.3)	2.8 (2.7–3.3)	0.387
Peripheral blood				
Leukocyte count (×10^9^/L)	10.1 (7.8–12.6)	10.3 (7.8–12.6)	9.2 (7.8–12.6)	0.533
C-reactive protein (mg/L)	9 (5–22)	9 (5–21)	15 (5–27)	0.149
Procalcitonin (ng/mL)	0.03 (0.01–0.06)	0.03 (0.01–0.06)	0.03 (0.01–0.08)	0.492

Data are median (interquartile range) or number (%) of patients. Abbreviations: CSF, cerebrospinal fluid; ID, infectious diseases; LP, lumbar puncture; TBE, tick-borne encephalitis. ^1^ Due to multiple statistical tests *p* < 0.001 was considered significant. ^2^ Criteria for undergoing head imaging prior to lumbar puncture [2,15]. Some patients had more than one neurological sign.

**Table 3 jcm-14-00045-t003:** Number (%) of patients with tick-borne encephalitis with computed tomography of the head performed before lumbar puncture according to selected indications by admitting service, i.e., infectious diseases or neurology.

Indication for Head Imaging ^1^	All *n* = 318	ID *n* = 256	Neurology *n* = 62	*p*-Value ^2^
Papilledema	/	/	/	/
Impaired consciousness ^3^	21/57 (36.8)	5/36 (13.9)	16/21 (76.2)	**<0.001**
Focal neurological signs	32/167 (19.2)	7/128 (5.5)	25/39 (64.1)	**<0.001**
Focal neurological signs without tremor	13/26 (50.0)	2/12 (16.7)	11/14 (78.6)	0.006
New onset seizure	2/2 (100.0)	1/1 (100.0)	1/1 (100.0)	>0.999
Glasgow coma scale of <10	2/5 (40.0)	1/4 (25.0)	1/1 (100.0)	0.400
Severely immunocompromised state ^4^	1/2 (50.0)	0	1/2 (50.0)	>0.999
Any of the above indications	39/183 (21.3)	11/140 (7.9)	28/43 (65.1)	**<0.001**
Without clear indication	18/135 (13.3)	6/116 (5.2)	12/19 (63.2)	**<0.001**
Any of the above or without clear indication	57 (17.9)	17 (6.6)	40 (64.5)	**<0.001**

Abbreviations: ID, infectious diseases. ^1^ Indications for head imaging before lumbar puncture according to Venkatesan and van de Beek [2,15]. ^2^ *p* values have only a descriptive purpose. ^3^ Qualitative or quantitative impairment. ^4^ Treatment with immunosuppressive medications.

**Table 4 jcm-14-00045-t004:** Factors associated with the decision to perform head imaging before lumbar puncture in patients with tick-borne encephalitis.

	Head Imaging Conducted Before Lumbar Puncture (Yes vs. No)	
	OR (95% CI)	*p* Value
Admitted in neurology vs. ID	22.0 (10.5–48.5)	<0.001
Focal neurological signs without tremor	0.62 (0.12–3.11)	0.569
Tremor	0.40 (0.15–1.02)	0.060
≥3 TBE characteristics ^1^	0.63 (0.25–1.64)	0.334
Clinical severity score of TBE ^2^	1.07 (0.95–1.22)	0.263

Abbreviations: OR, odds ratio; CI; confidence interval; ID, infectious diseases; TBE, tick-borne encephalitis. ^1^ Recent tick bite, biphasic course, fever, headache, nausea/vomiting, nuchal rigidity, tremor. ^2^ Assessed at hospital admission.

**Table 5 jcm-14-00045-t005:** Number (%) of patients with tick-borne encephalitis with selected microbiologic testing other than testing for tick-borne encephalitis virus by admitting service, i.e., infectious diseases or neurology.

Microbiologic Testing	All *n* = 318	ID *n* = 256	Neurology *n* = 62	*p* Value ^1^
Serology for Lyme borreliae	315 (99.1)	256 (100)	59 (95.2)	0.007
Herpes simplex virus 1/2 PCR in CSF	63 (19.8)	31 (12.1)	32 (51.6)	**<0.001**
Varicella-zoster virus PCR in CSF	55 (17.3)	25 (9.8)	30 (48.4)	**<0.001**
Enterovirus PCR in CSF	22 (6.9)	7 (2.7)	15 (24.2)	**<0.001**
PCR for EBV and/or CMV in CSF	22 (6.9)	5 (2.0)	17 (27.4)	**<0.001**
Bacterial culture of CSF	22 (6.9)	14 (5.5)	8 (12.9)	0.050
Listeria PCR in CSF	9 (2.8)	2 (0.8)	7 (11.3)	**<0.001**
Cryptococcal antigen in CSF	2 (0.6)	1 (0.4)	1 (1.6)	0.352
Blood cultures	28 (8.8)	23 (9.0)	5 (8.1)	1.000
Other ^2^	29 (9.1)	10 (3.9)	19 (30.6)	**<0.001**
Any microbiologic test ^3^	73 (23.0)	41 (16.0)	32 (51.6)	**<0.001**

Abbreviations: ID, infectious diseases; CMV, cytomegalovirus; CSF, cerebrospinal fluid; EBV, Epstein-Barr virus; ID, infectious diseases; PCR, polymerase chain reaction. ^1^ *p* values have only descriptive purpose. ^2^ Such as syphilis serology, human immunodeficiency virus serology and/or PCR, human herpesvirus 6 PCR, staining, culture and PCR for *Mycobacteria tuberculosis*. ^3^ Other than testing for tick-borne encephalitis virus and Lyme borreliae.

**Table 6 jcm-14-00045-t006:** Factors associated with worse clinical outcome/incomplete recovery at hospital discharge (modified Rankin score ≥ 2 vs. ≤ 1) in patients with tick-borne encephalitis according to the admitting service, i.e., neurology vs. infectious diseases.

	Clinical Outcome at Discharge from Hospital	
	OR (95% CI)	*p* Value
Admitted in neurology vs. ID	0.02 (−0.57–0.63)	0.949
Age	0.00 (−0.02–0.03)	0.876
Sex (male vs. female)	0.31 (−0.16–0.79)	0.197
Charlson comorbidity index	0.04 (−0.23–0.32)	0.788
Vaccinated (yes vs. no)	0.08 (−0.77–0.98)	0.861
Clinical severity score of TBE ^1^	−0.07 (−0.14–−0.02)	**0.015**

Abbreviations: OR, odds ratio; CI; confidence interval; ID, infectious diseases; TBE, tick-borne encephalitis. ^1^ Assessed at hospital discharge.

## Data Availability

All anonymous data generated or analyzed during this study are available from the corresponding author upon reasonable request (dasa.stupica@kclj.si).

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
