# Peer review of "Presentation, Management, and Outcome of Tick-Borne Encephalitis in Patients Referred to Infectious Diseases or Neurology†"

_jcm, 2024, doi:10.3390/jcm14010045_

Round 1
Reviewer 1 Report
Comments and Suggestions for Authors
The study assesses potential differences in the clinical presentation at admission and diagnostic and therapeutic management of patients with TBE initially referred to ID or neurology. The following comments require attention:
Abstract
The conclusion could better emphasize the clinical implications of the findings. Reframe the conclusion to underscore actionable insights.
Introduction
A more structured flow is needed to connect the background, study gap, and objectives. Use subheadings or transitional phrases to enhance flow.
Methods
The description of the clinical severity score is dense and might confuse readers. This section only briefly discusses the limitations of the retrospective design. Highlight retrospective design limitations briefly within the methods section.
Results
Some results (e.g., descriptions of clinical severity scores) are repeated unnecessarily across the text and figures. There is a lack of emphasis on the clinical significance of the findings. Discuss the practical implications of the findings alongside the statistical results.
Discussion
There are limited suggestions for how the findings could inform clinical practice or future research.
Need to be focused on evidence-based interpretations and avoid overgeneralizations.
Provide specific recommendations for improving clinical guidelines or management practices.
Conclusion
The article fails to adequately emphasize the broader implications or next steps. Need to expand on how the study’s findings could guide future research or policy changes.
Comments on the Quality of English LanguageThe language of the manuscript seems fine for me.
Author Response
Comment 1: The study assesses potential differences in the clinical presentation at admission and diagnostic and therapeutic management of patients with TBE initially referred to ID or neurology. The following comments require attention:
Response 1: We are grateful to the reviewer for his/her thoughtful critiques and suggestions that helped us make appropriate modifications and improve the manuscript.
Comment 2: Abstract
The conclusion could better emphasize the clinical implications of the findings. Reframe the conclusion to underscore actionable insights.
Response 2: Corrections made. Please see Abstract, lines 38‒45.
Comment 3: Introduction
A more structured flow is needed to connect the background, study gap, and objectives. Use subheadings or transitional phrases to enhance flow.
Response 3: Corrections made. Please see Introduction, lines 49‒85.
Comment 4: Methods
The description of the clinical severity score is dense and might confuse readers. This section only briefly discusses the limitations of the retrospective design. Highlight retrospective design limitations briefly within the methods section.
Response 4: Corrections made. Please see lines 107‒111 and lines 100‒101.
Comment 5: Results
Some results (e.g., descriptions of clinical severity scores) are repeated unnecessarily across the text and figures. There is a lack of emphasis on the clinical significance of the findings. Discuss the practical implications of the findings alongside the statistical results.
Response 5: Corrections made. Please see lines 208‒217. Practical implications of the findings are provided in the Discussion.
Comment 6: Discussion
There are limited suggestions for how the findings could inform clinical practice or future research.
Need to be focused on evidence-based interpretations and avoid overgeneralizations.
Provide specific recommendations for improving clinical guidelines or management practices.
Response 6: Corrections made. Please see lines 418‒426.
Comment 7: Conclusion
The article fails to adequately emphasize the broader implications or next steps. Need to expand on how the study’s findings could guide future research or policy changes.
Response 7: Corrections made. Due to the limitations of our study design, we believe we should refrain from making recommendations on policy changes. Rather we suggest areas of potential change in clinical management that could be further explored. Future research using more robust designs could confirm or reject these suggestions. Please see lines 465‒470.

Reviewer 2 Report
Comments and Suggestions for Authors
The study by Gulin et al. investigated the outcome of tick-birne encephalitis in patients referred to infections diseases or neurology. The results showed that there is no clear association among tested parameters. The reviewer feel that the initial design and the motivation of this study are questionable. There should be clear guidelines for doctors and physicians to suggest necessary diagnosis of patients who have specific symptoms. In this sense, one should not expect any correlations among those parameters. However, it should also be careful that the sample size can have significant impact on the results and conclusions. Despite those weakness, this study might be interesting for those who are involved in the optimzation of the hospital management and descision making. The following aspects are listed for authors to improve their manuscript and study.
1) The introduction needs significant care for better presentarion of the logic, motivation, and significance of this study. What are expcted and how we can process and use the results from this study.
2) The sample size might be another issue along with the specific time period (Covid) and location. The author should disscuss those aspects and clearly state the limitation of this study.
3) In terms of the parameters used, the authors also need to include the PMS of the patients and of their family.
4) Discussion section needs significant improvement with clear subsections.
Comments on the Quality of English LanguageMajor
Author Response
Comment 1: The study by Gulin et al. investigated the outcome of tick-birne encephalitis in patients referred to infections diseases or neurology. The results showed that there is no clear association among tested parameters. The reviewer feel that the initial design and the motivation of this study are questionable. There should be clear guidelines for doctors and physicians to suggest necessary diagnosis of patients who have specific symptoms. In this sense, one should not expect any correlations among those parameters. However, it should also be careful that the sample size can have significant impact on the results and conclusions. Despite those weakness, this study might be interesting for those who are involved in the optimzation of the hospital management and descision making. The following aspects are listed for authors to improve their manuscript and study.
Response 1: We are grateful to the reviewer for his/her thoughtful critiques and suggestions that helped us make appropriate modifications and improve the manuscript.
Comment 2: The introduction needs significant care for better presentarion of the logic, motivation, and significance of this study. What are expcted and how we can process and use the results from this study.
Response 2: Corrections made. Please see the Introduction.
Comment 3: The sample size might be another issue along with the specific time period (Covid) and location. The author should disscuss those aspects and clearly state the limitation of this study.
Response 3: Corrections made. Please see line 449‒450 and lines 454‒455.
Comment 4: In terms of the parameters used, the authors also need to include the PMS of the patients and of their family.
Response 4: We apologize, but we do not understand what the reviewer means by PMS. Post-marketing surveillance, progressive multiple sclerosis, Phelan-McDermid Syndrome, and premenstrual syndrome would all correspond to the acronym PMS. None of these entities seem to relate to this manuscript.
Comment 5: Discussion section needs significant improvement with clear subsections.
Response 5: We believe the Discussion is already divided into subsections following the same sequence as that of the Results section.
Reviewer 3 Report
Comments and Suggestions for Authors
This study presents data from Slovenia regarding tick-borne encephalitis and its management. Some points for a major revision are below:
The Introduction is very very brief. Please expand. For example, authors can present data not necessarily focused on this specific disorder, but generally on the management and the involvement not only of healthcare professionals, but also of family members on the dissemination of knowledge about the diagnosis and treatment/rehabilitation of the disease (for a relevant discussion on this: https://jamanetwork.com/journals/jamaneurology/article-abstract/585140). This general approach can also be discussed in the Discussion section. Furthermore, the differences in the healthcare systems and the cross-cultural differences must also be highlighted especially for the field of neurology and how healthcare professionals should organize their approach towards these disorders in different healthcare systems/countries/sociocultural contexts (https://www.neurology.org/doi/10.1212/WNL.0000000000004731?url_ver=Z39.88-2003&rfr_id=ori:rid:crossref.org&rfr_dat=cr_pub%20%200pubmed and https://journals.lww.com/neurotodayonline/fulltext/2006/09190/how_do_other_healthcare_systems_work__neurologists.14.aspx).
In addition, authors should think about more complex statistics (e.g. regressions).
Please explain all figures in the main text.
Author Response
Comment 1: This study presents data from Slovenia regarding tick-borne encephalitis and its management. Some points for a major revision are below:
Response 1: We are grateful to the reviewer for his/her thoughtful critiques and suggestions that helped us make appropriate modifications and improve the manuscript.
Comment 2: The Introduction is very very brief. Please expand. For example, authors can present data not necessarily focused on this specific disorder, but generally on the management and the involvement not only of healthcare professionals, but also of family members on the dissemination of knowledge about the diagnosis and treatment/rehabilitation of the disease (for a relevant discussion on this: https://jamanetwork.com/journals/jamaneurology/article-abstract/585140). This general approach can also be discussed in the Discussion section. Furthermore, the differences in the healthcare systems and the cross-cultural differences must also be highlighted especially for the field of neurology and how healthcare professionals should organize their approach towards these disorders in different healthcare systems/countries/sociocultural contexts (https://www.neurology.org/doi/10.1212/WNL.0000000000004731?url_ver=Z39.88-2003&rfr_id=ori:rid:crossref.org&rfr_dat=cr_pub%20%200pubmed and https://journals.lww.com/neurotodayonline/fulltext/2006/09190/how_do_other_healthcare_systems_work__neurologists.14.aspx).
Response 2: Corrections made. Please see the Introduction.
Comment 3: In addition, authors should think about more complex statistics (e.g. regressions).
Response 3: As suggested by the reviewer, we performed additional logistic regression to assess the association between admitting service (neurology vs ID) and physicians’ decision to perform head CT before LP adjusted for selected clinical characteristics which might influence this decision. Please see Table 4.
Comment 4: Please explain all figures in the main text.
Response 4: Corrections made. Please see lines 107‒111 and lines 296‒301. In response to reviewer 1, we deleted Figure 2 to avoid unnecessary repetition across text and figures.
Round 2
Reviewer 2 Report
Comments and Suggestions for Authors
Sorry for the confusing abbreviation. This reviewer suggests the authors should also consider the medical history of the patients used for analysis.
Author Response
Comment 1: Sorry for the confusing abbreviation. This reviewer suggests the authors should also consider the medical history of the patients used for analysis.
Response 1: Corrections done. Please see lines 92˗93 of the attached revised manuscript.

Reviewer 3 Report
Comments and Suggestions for Authors
Most points raised by the reviewers have been answered in this revision.
Author Response
Comment 1: Most points raised by the reviewers have been answered in this revision.
Response 1: We thank the reviewer for his/her comment.